# When Does Differentially Private Learning Not Suffer in High Dimensions?

**Xuechen Li**[*]
Stanford University
lxuechen@cs.stanford.edu

**Daogao Liu**[*]
University of Washington
dgliu@uw.edu

**Tatsunori Hashimoto**
Stanford University
thashim@stanford.edu

**Huseyin A. Inan**
Microsoft Research
huseyin.inan@microsoft.com

**Janardhan Kulkarni**
Microsoft Research
jakul@microsoft.com

**Yin Tat Lee**
University of Washington
Microsoft Research
yintat@uw.edu

**Abhradeep Guha Thakurta**
Google Research
athakurta@google.com

## Abstract

Large pretrained models can be fine-tuned with differential privacy to achieve performance approaching that of non-private models. A common theme in these results is the surprising observation that high-dimensional models can achieve favorable privacy-utility trade-offs. This seemingly contradicts known results on the model-size dependence of differentially private convex learning and raises the following research question: When does the performance of differentially private learning not degrade with increasing model size? We identify that the magnitudes of gradients projected onto subspaces is a key factor that determines performance. To precisely characterize this for private convex learning, we introduce a condition on the objective that we term *restricted Lipschitz continuity* and derive improved bounds for the excess empirical and population risks that are dimension-independent under additional conditions. We empirically show that in private fine-tuning of large language models, gradients obtained during fine-tuning are mostly controlled by a few principal components. This behavior is similar to conditions under which we obtain dimension-independent bounds in convex settings. Our theoretical and empirical results together provide a possible explanation for the recent success of large-scale private fine-tuning. Code to reproduce our results can be found at https://github.com/lxuechen/private-transformers/tree/main/examples/classification/spectral_analysis.

## 1 Introduction

Recent works have shown that large publicly pretrained models can be differentially privately fine-tuned on small downstream datasets with performance approaching those attained by non-private models. In particular, past works showed that pretrained BERT [DCLT18] and GPT-2 [RNSS18, RWC[+]19] models can be fine-tuned to perform well for text classification and generation under a privacy budget of $\varepsilon \in [2, 6]$ [LTLH21, YNB[+]21]. More recently, it was shown that pretrained

---

[*]Equal contribution

36th Conference on Neural Information Processing Systems (NeurIPS 2022).

ResNets [HZRS16] and vision-Transformers [DBK$^+$20] can be fine-tuned to perform well for ImageNet classification under single digit privacy budgets [DBH$^+$22, MTKC22].

One key ingredient in these successes has been the use of large pretrained models with millions to billions of parameters. These works generally highlighted the importance of two phenomena: (i) large pretrained models tend to experience good privacy-utility trade-offs when fine-tuned, and (ii) the trade-off improves with the improvement of the quality of the pretrained model (correlated with increase in size). While the power of scale and pretraining have been demonstrated numerous times in non-private deep learning [KMH$^+$20], one common wisdom in private learning had been that large models tend to perform worse. This intuition was based on (a) results in differentially private convex optimization, most of which predicted that errors would scale proportionally with the dimension of the learning problem in the worst case, and (b) empirical observations that the noise injected to ensure privacy tends to greatly exceed the gradient in magnitude for large models [YZCL21a, Kam20].

For instance, consider the problem of differentially private convex *empirical risk minimization* (ERM). Here, we are given a dataset of $n$ examples $\mathcal{D} = \{s_j\}_{j=1}^n \in \mathcal{S}^n$, a convex set $\mathcal{K} \subseteq \mathbb{R}^d$ (not necessarily bounded), and the goal is to perform the optimization

$$\text{minimize}_{x \in \mathcal{K}} F(x; \mathcal{D}) = \frac{1}{n} \sum_{j=1}^n f(x; s_j)$$

subject to differential privacy, where $f(\cdot; s)$ is convex over $\mathcal{K}$ for all $s \in \mathcal{S}$. For bounded $\mathcal{K}$, past works presented matching upper and lower bounds that are dimension-dependent under the usual Lipschitz assumption on the objective [BST14, CMS11]. These results seem to suggest that the performance of differentially private ERM algorithms inevitably degrades with increasing problem size in the worst case, and present a seeming discrepancy between recent empirical results on large-scale fine-tuning.[2]

To better understand the relation between problem size and the performance of differentially private learning, we study the following question both theoretically and empirically:

*When does the performance of differentially private stochastic gradient descent (DP-SGD) not degrade with increasing problem dimension?*

On the theoretical front, we show that DP-SGD can result in dimension-independent error bounds even when gradients span the entire ambient space for unconstrained optimization problems. We identify that the standard dependence on the dimension of the ambient space can be replaced by the magnitudes of gradients projected onto subspaces of varying dimensions. We formalize this in a condition that we call *restricted Lipschitz continuity* and derive refined bounds for the excess empirical and population risks for DP-SGD when loss functions obey this condition. We show that when the restricted Lipschitz coefficients decay rapidly, both the excess empirical and population risks become dimension-independent. This extends a previous work which derived rank-dependent bounds for learning generalized linear models in an unconstrained space [SSTT21].

Our theoretical results shed light on the recent success of large-scale differentially private fine-tuning. We empirically show that gradients of language models during fine-tuning are mostly controlled by a few principal components — a behavior that is similar to conditions under which we obtain dimension-independent bounds for private convex ERM. This provides a possible explanation for the observation that densely fine-tuning with DP-SGD need not necessarily experience much worse performance than sparsely fine-tuning [LTLH21]. Moreover, it suggests that DP-SGD can be adaptive to problems that are effectively low-dimensional (as characterized by restricted Lipschitz continuity) without further algorithmic intervention.

We summarize our contributions below.

(1) We introduce a condition on the objective function that we term restricted Lipschitz continuity. This condition generalizes the usual Lipschitz continuity notion and gives rise to refined analyses when magnitudes of gradients projected onto diminishing subspaces decay rapidly.

(2) Under restricted Lipschitz continuity, we present refined bounds on the excess empirical and population risks for DP-SGD when optimizing convex objectives. These bounds generalize

---

[2]We judiciously choose to describe the discrepancy as seeming, since the refined analysis presented in the current work suggests that the discrepancy is likely non-existent.

previous dimension-independent results [SSTT21] and are broadly applicable to cases where gradients are full rank but most coordinates only marginally influence the objective.

(3) Our theory sheds light on recent successes of large-scale differentially private fine-tuning of language models. We show that gradients obtained through fine-tuning mostly lie in a subspace spanned by a few principal components — a behavior similar to when optimizing a restricted Lipschitz continuous loss with decaying coefficients. These empirical results provide a possible explanation for the recent success of large-scale private fine-tuning.

## 2 Preliminaries

We define the notation used throughout this work and state the problems of differentially private empirical risk minimization and differentially private stochastic convex optimization. Finally, we give a brief recap of differentially private stochastic gradient descent, and existing dimension-dependent and dimension-independent results in the literature.

**Notation & Terminology.** For a positive integer $n \in \mathbb{N}_+$, define the shorthand $[n] = \{1, \ldots, n\}$. For a vector $x \in \mathbb{R}^d$, denote its $\ell_2$-norm by $\|x\|_2$. Given a symmetric $M \in \mathbb{R}^{d \times d}$, let $\lambda_1(M) \geq \lambda_2(M) \geq \cdots \geq \lambda_d(M)$ denote its eigenvalues. Given a positive semidefinite matrix $A$, let $\|x\|_A = (x^\top A x)^{1/2}$ denote the induced Mahalanobis norm. For scalar functions $f$ and $g$, we write $f \lesssim g$ if there exists a positive constant $C$ such that $f(x) \leq Cg(x)$ for all input $x$ in the domain.

### 2.1 Differentially Private Empirical Risk Minimization and Stochastic Convex Optimization

Before stating the theoretical problem of interest, we recall the basic concepts of Lipschitz continuity, convexity, and approximate differential privacy.

**Definition 2.1** (Lipschitz Continuity). The loss function $h : \mathcal{K} \to \mathbb{R}$ is $G$-Lipschitz with respect to the $\ell_2$ norm if for all $x, x' \in \mathcal{K}$, $|f(x) - f(x')| \leq G\|x - x'\|_2$.

**Definition 2.2** (Convexity). The loss function $h : \mathcal{K} \to \mathbb{R}$ is convex if $h(\alpha x + (1 - \alpha)y) \leq \alpha h(x) + (1 - \alpha)h(y)$, for all $\alpha \in [0, 1]$, and $x, y$ in a convex domain $\mathcal{K}$.

**Definition 2.3** (Approximate Differential Privacy [DR+14]). A randomized algorithm $\mathcal{M}$ is $(\varepsilon, \delta)$-differentially private if for all neighboring datasets $\mathcal{D}$ and $\mathcal{D}'$ that differ by a single record and all sets $\mathcal{O} \subset \mathrm{range}(\mathcal{M})$, the following expression holds

$$\Pr[\mathcal{M}(\mathcal{D}) \in \mathcal{O}] \leq \exp(\varepsilon)\Pr[\mathcal{M}(\mathcal{D}') \in \mathcal{O}] + \delta.$$

In this work, we study both *differentially private empirical risk minimization* (DP-ERM) for convex objectives and *differentially private stochastic convex optimization* (DP-SCO).

In DP-ERM for convex objectives, we are given a dataset $\mathcal{D} = \{s_j\}_{j \in [n]} \in \mathcal{S}^n$ of $n$ examples. Each per-example loss $f(\cdot; s_j)$ is convex over the convex body $\mathcal{K} \subseteq \mathbb{R}^d$ and $G$-Lipschitz. We aim to design an $(\varepsilon, \delta)$-DP algorithm that returns a solution $x^{\mathrm{priv}}$ which approximately minimizes the empirical risk $F(x; \mathcal{D}) := \frac{1}{n}\sum_{s_j \in \mathcal{D}} f(x; s_j)$. The term $\mathbb{E}_{x^{\mathrm{priv}}}\left[F(x^{\mathrm{priv}}; \mathcal{D}) - \min_{x \in \mathcal{K}} F(x; \mathcal{D})\right]$ is referred to as the *excess empirical risk*.

In DP-SCO, we assume the per-example loss $f(\cdot; s)$ is convex and $G$-Lipschitz for all $s \in \mathcal{S}$, and we are given $n$ examples drawn i.i.d. from some (unknown) distribution $\mathcal{P}$. The goal is to design an $(\varepsilon, \delta)$-DP algorithm which approximately minimizes the population risk $F(x; \mathcal{P}) := \mathbb{E}_{s \sim \mathcal{P}}[f(x; s)]$. The term $\mathbb{E}_{x^{\mathrm{priv}}}\left[F(x^{\mathrm{priv}}; \mathcal{P}) - \min_{x \in \mathcal{K}} F(x; \mathcal{P})\right]$ is referred to as the *excess population risk*.

### 2.2 Differentially Private Stochastic Gradient Descent

*Differentially Private Stochastic Gradient Descent* (DP-SGD) [ACG+16, SCS13] is a popular algorithm for DP convex optimization. For the theoretical analysis, we study DP-SGD for *unconstrained* optimization. To facilitate analysis, we work with the $\ell_2$ regularized objective expressed as

$$F_\alpha(x; \mathcal{D}) = \frac{1}{n}\sum_{j=1}^n f(x; s_j) + \frac{\alpha}{2}\|x - x^{(0)}\|_2^2.$$

To optimize this objective, DP-SGD independently samples an example in each iteration and updates the solution by combining the gradient with an isotropic Gaussian whose scale is proportional to $G$, the Lipschitz constant of $f$. Algorithm 1 presents the pseudocode.

---

**Algorithm 1:** DP-SGD for optimizing regularized finite-sum objectives

---

1 **Input:** Initial iterate $x^{(0)}$, dataset $\mathcal{D} = \{s_j\}_{j \in [n]}$, per-step noise magnitude $\sigma$, number of updates $T$, learning rate $\eta$, Lipschitz constant $G$ of $f$.
2 **for** $t = 1, \ldots, T$ **do**
3 $\quad$ $j_t \sim \text{Uniform}([n])$
4 $\quad$ $x^{(t)} = x^{(t-1)} - \eta\Big(\nabla f(x^{(t-1)}; s_{j_t}) + \alpha(x^{(t-1)} - x^{(0)}) + G \cdot \zeta_t\Big), \quad \zeta_t \sim \mathcal{N}(0, \sigma^2 I_d)$
5 **end**
6 **Return:** $\overline{x} \stackrel{\text{def}}{=} \frac{1}{T} \sum_{t=1}^{T} x^{(t)}$.

---

It is straightforward to show that Algorithm 1 satisfies differential privacy. The following lemma quantifies the overall privacy spending and builds on a long line of work on accounting the privacy loss of compositions [ACG+16, BBG18].

**Lemma 2.4** ([KLL21]). *There exists constants $c_1$ and $c_2$ such that for $n \geq 10$, $\varepsilon < c_1 T/n^2$ and $\delta \in (0, \frac{1}{2}]$, DP-SGD (Algorithm 1) is $(\varepsilon, \delta)$-DP whenever $\sigma \geq \frac{c_2 \sqrt{T \log(1/\delta)}}{\varepsilon n}$.*

### 2.3 On the Dimension Dependence of Private Learning

Early works on bounding the excess empirical and population risks for privately optimizing convex objectives focused on a constrained optimization setup where algorithms typically project iterates back onto a fixed bounded domain after each noisy gradient update. Results in this setting suggested that risks are inevitably dimension-dependent in the worst case. For instance, it was shown that the excess empirical risk bound $\Theta(G \|\mathcal{K}\|_2 \sqrt{d \log(1/\delta)} n^{-1} \varepsilon^{-1})$ and excess population risk bound $\Theta(G \|\mathcal{K}\|_2 (n^{-1/2} + \sqrt{d \log(1/\delta)} n^{-1} \varepsilon^{-1}))$ are tight when privately optimizing convex $G$-Lipschitz objectives in a convex domain of diameter $\|\mathcal{K}\|_2$ [BST14]. Moreover, the lower bound instances in these works imply that such dimension-dependent lower bounds also apply when one considers the class of loss functions whose gradients are low-rank.

The body of work on unconstrained convex optimization is arguably less abundant, with the notable result that differentially private gradient descent need not suffer from a dimension-dependent penalty when learning generalized linear models with low-rank data (equivalently stated, when gradients are low-rank) [SSTT21]. Our main theoretical results (Theorems 3.3 and 3.5) extend this line of work and show that dimension-independence is achievable under weaker conditions.

## 3 Refined Dimension-Dependence via Restricted Lipschitz Continuity

In this section, we formally introduce the restricted Lipschitz continuity condition and derive improved bounds for the excess empirical and population risks for DP-SGD when optimizing convex objectives.

**Definition 3.1** (Restricted Lipschitz Continuity). We say that the loss function $h : \mathcal{K} \to \mathbb{R}$ is restricted Lipschitz continuous with coefficients $G_0, \ldots, G_d$, if for all $k \in [d]$, there exists an orthogonal projection matrix $P_k$ with rank $k$ such that

$$\|(I - P_k)\nabla h(x)\|_2 \leq G_k,$$

for all $x \in \mathcal{K}$ and all subgradients $\nabla h(x) \in \partial h(x)$.

Note that any $G$-Lipschitz function is also trivially restricted Lipschitz continuous with coefficients $G = G_0 = G_1 = \cdots = G_d$, since orthogonal projections never increase the $\ell_2$-norm of a vector (generally, it is easy to see that $G = G_0 \geq G_1 \geq G_2 \geq \cdots \geq G_d = 0$). On the other hand, we expect that a function which exhibits little growth in subspaces of dimension $k$ to have a restricted Lipschitz coefficient $G_{d-k}$ of almost 0.

Our bounds on DP convex optimization will characterize errors through the use of restricted Lipschitz coefficients. We now summarize the main assumptions before presenting the core theoretical results.

**Assumption 3.2.** *The per-example loss function $f(\cdot; s)$ is convex and $G$-Lipschitz continuous for all $s \in \mathcal{S}$. The average loss $F(\cdot; \mathcal{D})$ is restricted Lipschitz continuous with coefficients $\{G_k\}_{k=1}^d$.*

## 3.1 Bounds for Excess Empirical Loss

We present the main theoretical result on DP-ERM for convex objectives. The result consists of two pieces: Equation (1) is a general bound that is applicable to any sequence of restricted Lipschitz coefficients; Equation (2) specializes the previous bound when the sequence of coefficients decays rapidly and demonstrates dimension-independent error scaling.

**Theorem 3.3** (Excess Empirical Loss). *Let $\delta \in (0, \frac{1}{2}]$, and $\varepsilon \in (0, 10]$. Under Assumption 3.2, for any positive integer $k \in [d]$, setting $T = \Theta(n^2 + d\log^2 d)$, $\sigma = \Theta\left(\frac{\sqrt{T\log(1/\delta)}}{n\varepsilon}\right)$, $\eta = \sqrt{\frac{D^2}{T \cdot G_0^2 \cdot k\sigma^2}}$ and $\alpha = \frac{1}{D}\sqrt{\sum_{s=1}^S s^2 2^s G_{2^{s-1}k}^2}$, where $S = \lfloor \log(d/k) \rfloor + 1$, DP-SGD (Algorithm 1) is $(\varepsilon, \delta)$-DP, and*

$$\mathbb{E}\left[F(\overline{x}; \mathcal{D}) - \min_x F(x; \mathcal{D})\right] \lesssim \frac{G_0 D \sqrt{k\log(1/\delta)}}{\varepsilon n} + D\sqrt{\sum_{s=1}^S s^2 2^s G_{2^{s-1}k}^2}, \qquad (1)$$

*where $\|x^{(0)} - \arg\min_x F(x; \mathcal{D})\|_2 \le D$, $\overline{x}$ is the (random) output of DP-SGD (Algorithm 1), and the expectation is over the randomness of $\overline{x}$. Moreover, if for some $c > 1/2$, we have $G_k \le G_0 k^{-c}$ for all $k \in [d]$, and in addition $n \ge \varepsilon^{-1}\sqrt{\log(1/\delta)}$, minimizing the right hand side of (1) with respect to $k$ yields*

$$\mathbb{E}\left[F(\overline{x}; \mathcal{D}) - \min_x F(x; \mathcal{D})\right] \lesssim G_0 D \cdot \left(\frac{\sqrt{\log(1/\delta)}}{\varepsilon n}\right)^{2c/(1+2c)}. \qquad (2)$$

We include a sketch of the proof techniques in Section 3.3 and defer the full proof to Appendix A.

*Remark* 3.4. Consider DP-ERM for learning generalized linear models with convex and Lipschitz losses. When the (empirical) data covariance is of rank $r < d$, the span of gradients $\text{span}(\{\nabla_x F(x)\})$ is also of rank $r$. Thus, the average loss is restricted Lipschitz continuous with coefficients where $G_{r'} = 0$ for all $r' > r$. Setting $k = r$ in (1) yields the excess empirical risk bound of order $O\left(G_0 D\sqrt{r \cdot \log(1/\delta)}\varepsilon^{-1}n^{-1}\right)$. This recovers the previous dimension-independent result [SSTT21].

The restricted Lipschitz continuity condition can be viewed as a generalized notion of rank. The result captured in (2) suggests that the empirical loss achieved by DP-SGD does not depend on the problem dimension if the sequence of restricted Lipschitz coefficients decays rapidly. We leverage these insights to build intuition for understanding private fine-tuning of language model in Section 4.

## 3.2 Bounds for Excess Population Loss

For DP-SCO, we make use of the *stability* of DP-SGD to bound its generalization error [BE02], following previous works [BFTT19, BFGT20, SSTT21]. The bound on the excess population loss follows from combining the bounds on the excess empirical risk and the generalization error.

**Theorem 3.5** (Excess Population Loss). *Let $\delta \in (0, \frac{1}{2}]$, and $\varepsilon \in (0, 10]$. Under Assumption 3.2, for any positive integer $k \in [d]$, by setting $T = \Theta(n^2 + d\log^2 d)$, $\sigma = \Theta\left(\frac{\sqrt{T\log(1/\delta)}}{n\varepsilon}\right)$, $\eta = \sqrt{\frac{D^2}{T \cdot G_0^2(T/n + k\sigma^2)}}$ and $\alpha = \frac{1}{D}\sqrt{\sum_{s=1}^S s^2 2^s G_{2^{s-1}k}^2}$, where $S = \lfloor \log(d/k) \rfloor + 1$, DP-SGD (Algorithm 1) is $(\varepsilon, \delta)$-DP, and*

$$\mathbb{E}\left[F(\overline{x}; \mathcal{P}) - \min_x F(x; \mathcal{P})\right] \lesssim \frac{G_0 D}{\sqrt{n}} + \frac{G_0 D\sqrt{k\log(1/\delta)}}{\varepsilon n} + D\sqrt{\sum_{s=1}^S s^2 2^s G_{2^{s-1}k}^2},$$

where $\|x^{(0)} - \arg\min_x F(x; \mathcal{P})\|_2 \leq D$, $\overline{x}$ is the (random) output of DP-SGD (Algorithm 1), and the expectation is over the randomness of $\overline{x}$.

Moreover, if for some $c > 1/2$, we have $G_k \leq G_0 k^{-c}$ for all $k \in [d]$, and in addition $n > \varepsilon^{-1}\sqrt{\log(1/\delta)}$, minimizing the above bound with respect to $k$ yields

$$\mathbb{E}\left[F(\overline{x}; \mathcal{P}) - \min_x F(x; \mathcal{P})\right] \lesssim \frac{G_0 D}{\sqrt{n}} + G_0 D \cdot \left(\frac{\sqrt{\log(1/\delta)}}{\varepsilon n}\right)^{2c/(1+2c)}.$$

*Remark* 3.6. Our result on DP-SCO also recovers the DP-SCO rank-dependent result for learning generalized linear models with convex and Lipschitz losses [SSTT21].

*Remark* 3.7. When $c > 1/2$, $\varepsilon = \Theta(1)$ and $\delta = 1/\text{poly}(n)$, the population loss matches the (non-private) informational-theoretical lower bound $\Omega(G_0 D/\sqrt{n})$ [AWBR09].

*Remark* 3.8. Our results on DP-ERM and DP-SCO naturally generalize to (full-batch) DP-GD.

### 3.3 Overview of Proof Techniques

The privacy guarantees in Theorems 3.3 and 3.5 follow from Lemma 2.4. It suffices to prove the utility guarantees. We give an outline of the main proof technique and defer full proofs to the supplementary. The following is a sketch of the core technique for deriving (2) in Theorem 3.3.

By convexity, the error term of SGD is upper bounded as follows

$$f_j(x^{(t)}) - f_j(x^*) \leq \nabla f_j(x^{(t)})^\top (x^{(t)} - x^*), \tag{3}$$

where $j \in [n]$ is the random index sampled at iteration $t$. By definition of $G_k$, we know that there is a $k$ dimensional subspace $U$ such that the gradient component orthogonal to $U$ is small when $G_k$ is small. Naïvely, one decomposes the gradient $\nabla f_j(x^{(t)}) = g_1 + g_2$, where $g_1 \in U$ and $g_2 \in U^\perp$, and separately bounds the two terms $g_1^\top (x^{(t)} - x^*)$ and $g_2^\top (x^{(t)} - x^*)$. Since $g_1$ lies in a $k$ dimensional subspace, one can follow existing arguments on DP-SGD to bound $g_1^\top (x^{(t)} - x^*)$. Unfortunately, this argument does not lead to dimension-independence. Although $\|g_2\|_2 \leq G_k$ (which can be small for large $k$), the term $\|x^{(t)} - x^*\|_2$ can be as large as $\Omega(\sqrt{d})$ with high probability due to the isotropic Gaussian noise injected in DP-SGD. Therefore, the naïve upper bound on $|g_2^\top (x^{(t)} - x)|$ can be as large $O(\sqrt{d}G_k)$.

Our key idea is to partition the whole space $\mathbb{R}^d$ into $\lfloor \log(d/k) \rfloor + 2$ orthogonal subspaces, expressing the error term $\nabla f_j(x^{(t)})^\top (x^{(t)} - x^*)$ as the sum of individual terms each of which corresponds to a projection to a particular subspace. Fix the positive integer $k \leq d$, and consider the following subspaces: Let $U_0 = \text{range}(P_k)$, $U_s$ be the subspace orthogonal to all previous subspaces such that $\bigoplus_{i=0}^s U_i \supseteq \text{range}(P_{2^s k})$ for $s = 1, 2, \cdots, \lfloor \log(d/k) \rfloor$, and $U_S$ be the subspace such that the orthogonal direct sum of all subspaces $\{U_i\}_{i=0}^S$ is $\mathbb{R}^d$ where $S = \lfloor \log(d/k) \rfloor + 1$. Here, $P_i$ is the orthogonal projection matrix with rank $i$ promised by $G_i$ in Assumption 3.2. Let $Q_s$ be the orthogonal projection to the subspace $U_s$. Observe that $\text{rank}(Q_s) \leq 2^s k$ and $\|Q_s \nabla F(x)\|_2 \leq G_{2^{s-1}k}$ for all $x$ and all $s \geq 1$. Rewriting the right hand side of (3) with this decomposition yields

$$f_j(x^{(t)}) - f_j(x^*) \leq \left(Q_0 \nabla f_j(x^{(t)}) + \sum_s Q_s \nabla f_j(x^{(t)})\right)^\top (x^{(t)} - x^*).$$

On the one hand, if $G_k$ decays rapidly, $\|\mathbb{E}_j[Q_s \nabla f_j]\|_2$ will be small for large $s$. On the other hand, we expect $\|Q_s(x^{(t)} - x^*)\|_2$ to be small for small $s$ where $Q_s$ is an orthogonal projection onto a small subspace. Thus, for each $s$, $\nabla f_j(x^{(t)})^\top Q_s(x^{(t)} - x^*)$ is small either due to a small gradient (small $Q_s \nabla f_j$ in expectation over the random index) or small noise (small $Q_s(x^{(t)} - x^*)$), since noise injected in DP-SGD is isotropic. More formally, in Lemma A.1, we show that for any projection matrix $Q$ with rank $r$, $\|Q(x^{(t)} - x^{(0)})\|_2$ can be upper bounded by a term that depends only on $r$ (rather than $d$).

## 4 Numerical Experiments

The aim of this section is twofold. In Section 4.1, we study a synthetic example that matches our theoretical assumptions and show that DP-SGD attains dimension-independent empirical and

population loss when the sequence of restricted Lipschitz coefficients decays rapidly—even when gradients span the entire ambient space. In Section 4.2, we study a stylized example of privately fine-tuning large language models. Building on the previous theory, we provide insights as to why dense fine-tuning can yield good performance.

## 4.1 Synthetic Example: Estimating the Generalized Geometric Median

We privately estimate the geometric median which minimizes the average Mahalanobis distance. Specifically, let $x_i \in \mathbb{R}^d$ for $i \in [n]$ be feature vectors drawn i.i.d. from some distribution $P_x$, each of which is treated as an individual record. Denote the entire dataset as $\mathcal{D} = \{x_i\}_{i=1}^n$. Subject to differential privacy, we perform the following optimization

$$\min_{x \in \mathbb{R}^d} F_\alpha(x) = \frac{1}{n} \sum_{i=1}^n f_i(x) + \frac{\alpha}{2} \left\| x - x^{(0)} \right\|_2^2 = \frac{1}{n} \sum_{i=1}^n \|x - x_i\|_A + \frac{\alpha}{2} \left\| x - x^{(0)} \right\|_2^2, \quad (4)$$

where we adopt the shorthand $f_i(x) = f(x; x_i) = \|x - x_i\|_A$. When $A = I_d$ and $\alpha = 0$ (without the regularization term), the problem reduces to estimating the usual geometric median (commonly known as center of mass).

For this example, individual gradients are bounded since $\|\nabla f_i(x)\|_2 = \|A(x - x_i)/\|x - x_i\|_A\|_2 \leq \lambda_1(A^{1/2}) = G_0$. More generally, the restricted Lipschitz coefficients of $F(x)$ are the eigenvalues of $A^{1/2}$, since

$$\|Q_k \nabla F(x)\|_2 = \left\| Q_k A^{1/2} \frac{1}{n} \sum_{i=1}^n \frac{A^{1/2}(x - x_i)}{\|x - x_i\|_A} \right\|_2 \leq \|Q_k A^{1/2}\|_{op} = \lambda_{k+1}(A^{1/2}) = G_k,$$

where $Q_k = I - P_k$ is chosen to be the rank $(d - k)$ orthogonal projection matrix that projects onto the subspace spanned by the bottom $(d - k)$ eigenvectors of $A^{1/2}$.

To verify our theory, we study the optimization and generalization performance of DP-SGD for minimizing (4) under Mahalanobis distances induced by different $A$ as the problem dimension grows. The optimization performance is measured by the final training error, and the generalization performance is measured by the population quantity $\mathbb{E}_{x \sim P_x, \bar{x}}[\|\bar{x} - x\|_A]$, where $\bar{x}$ denotes the random output of DP-SGD. We study the dimension scaling behavior for $A$ being one of

$$A_{\text{const}} = \text{diag}(1, \ldots, 1), \quad A_{\text{sqrt}} = \text{diag}(1, 1/\sqrt{2}, \ldots, 1/\sqrt{d}), \quad A_{\text{linear}} = \text{diag}(1, 1/2, \ldots, 1/d),$$

where $\text{diag} : \mathbb{R}^d \to \mathbb{R}^{d \times d}$ maps vectors onto square matrices with inputs on the diagonal. In all cases, the span of gradients $\text{span}(\{\nabla F(x)\})$ is the ambient space $\mathbb{R}^d$, since $A$ is of full rank. To ensure the distance from the initial iterate $\beta^{(0)} = 0$ to the optimum is the same for problem instances of different dimensions, we let feature vectors $\{x_i\}_{i=1}^n$ take zero values in any dimension $k > d_{\min}$, where $d_{\min}$ is the dimension of the smallest problem in our experiments. Our theoretical bounds suggest that when the sequence of restricted Lipschitz coefficients is constant (when $A = A_{\text{const}}$), the excess empirical loss grows with the problem dimension, whereas when the sequence of $k$th-Lipschitz constants rapidly decays with $k$ (when $A = A_{\text{sqrt}}$ or $A = A_{\text{linear}}$), the excess empirical loss does not grow beyond a certain problem dimension. Figure 1 empirically captures this phenomenon. We include additional experimental setup details in Appendix C.

## 4.2 Why Does Dense Fine-Tuning Work Well for Pretrained Language Models?

Stated informally, our bounds in Theorem 3.5 imply that DP-SGD obtains dimension-independent errors if gradients approximately reside in a subspace much smaller than the ambient space. Inspired by these results for the convex case, we now turn to study dense language model fine-tuning [LTLH21] and provide a possible explanation for their recent intriguing success — fine-tuning gigantic parameter vectors frequently results in moderate performance drops compared to non-private learning.

In the following, we present evidence that gradients obtained through fine-tuning mostly lie in a small subspace. We design subsequent experiments to work under a simplified setup. Specifically, we fine-tune DistilRoBERTa [SDCW19, LOG+19] under $\varepsilon = 8$ and $\delta = 1/n^{1.1}$ for sentiment classification on the SST-2 dataset [SPW+13]. We reformulate the label prediction problem as templated text prediction [LTLH21], and fine-tune only the query and value matrices in attention layers.

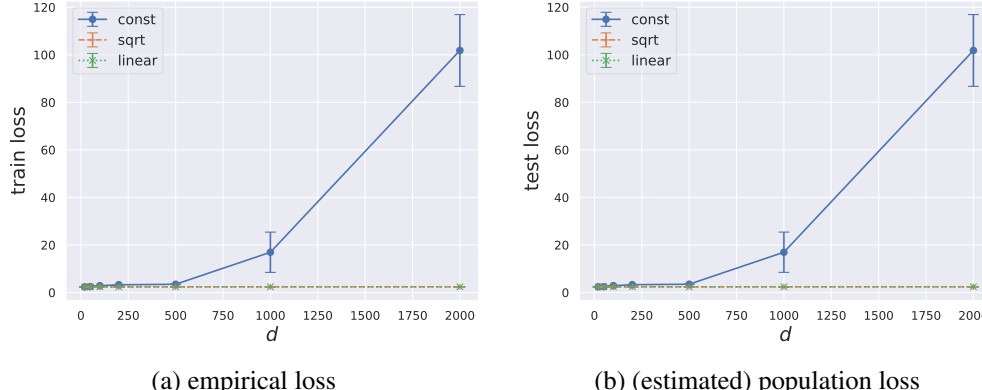

(a) empirical loss  (b) (estimated) population loss

Figure 1: The empirical and population losses grow with increasing problem dimension when the sequence of restricted Lipschitz coefficients remain constant. On the other hand, these losses remain almost constant when the sequence of restricted Lipschitz coefficients decays rapidly. Error bars represent one standard deviation over five runs of DP-SGD with the same hyperparameters which were tuned on separate validation data. For the same $A$, the optimal training error $\min_{x \in \mathbb{R}^d} F(x)$ is the same for problem instances with different dimensions (thus errors do not scale if learning was non-private). Each training run was performed with $\varepsilon = 2$, $\delta = 10^{-6}$, and $n = 10000$.

We focus on fine-tuning these specific parameter matrices due to the success of LoRA for non-private learning [HSW+21] which focuses on adapting the attention layers. Unlike LoRA, we fine-tune all parameters in these matrices rather than focusing on low-rank updates. This gives a setup that is lightweight enough to run spectral analyses computationally tractably but retains enough parameters ($\approx 7$ million) such that a problem of similar scale outside of fine-tuning results in substantial losses in utility.[3] For our setup, DP-SGD obtains a dev set accuracy approximately of $90\%$ and $92\%$, privately and non-privately, respectively. These numbers are similar to previous results obtained with the same pretrained model [YNB+21, LTLH21]. We include the full experimental protocol and additional results in Appendix D.

To provide evidence for the small subspace hypothesis, we sample gradients during fine-tuning and study their principal components. Specifically, we "over-train" by privately fine-tuning for $r = 2 \times 10^3$ updates and collect all the non-privatized average clipped gradients along the optimization trajectory. While fine-tuning for 200 and 2k updates have similar final dev set performance under our hyperparameters, the increased number of steps allows us to collect more gradients around the converged solution. This yields a gradient matrix $H \in \mathbb{R}^{r \times p}$, where $p \approx 7 \times 10^6$ is the size of the parameter vector. We perform PCA for $H$ with the orthogonal iteration algorithm [Dem97] and visualize the set of estimated singular values $\sigma_i(H) = \lambda_i(H^\top H)^{1/2}$ in terms of both (i) the density estimate, and (ii) their relation with the rank. Figure 2 (a) shows the top 1000 singular values sorted and plotted against their rank $k$ and the least squares fit on log-transformed inputs and outputs. The plot displays few large singular values which suggests that gradients are controlled through only a few principal directions. The linear fit suggests that singular values decay rapidly (at a rate of approximately $k^{-0.6}$).

To study the effects that different principal components have on fine-tuning performance, we further perform the following re-training experiment. Given the principal components, we privately re-fine-tune with gradients projected onto the top $k \in \{10, 20, 100\}$ components. Note that this projection applies only to the (non-privatized) average clipped gradients and the isotropic DP noise is still applied to all dimensions. Figure 2 (b) shows that the original performance can be attained by optimizing within a subspace of only dimension $k = 100$, suggesting that most of the dimensions of the 7 million parameter vector encode a limited learning signal.

While these empirical results present encouraging insights for the dimension-independent performance of fine-tuning, we acknowledge that this is not a complete validation of the restricted Lipschitz

---

[3]For instance, an off-the-shelf ResNet image classifier has 10 to 20+ million parameters. A plethora of works report large performance drops when training these models from scratch [YZCL21b, LWAFF21, DBH+22].

continuity condition and fast decay of coefficients (even locally near the optimum). We leave a more thorough analysis with additional model classes and fine-tuning tasks to future work.

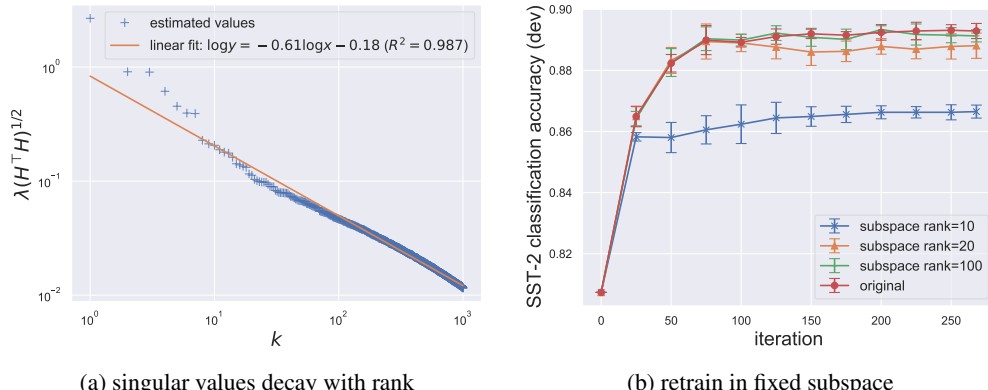

(a) singular values decay with rank

(b) retrain in fixed subspace

Figure 2: Gradients obtained through fine-tuning are controlled by a few principal components. *Left:* Singular values decay rapidly with their rank. *Right:* Retraining with gradients projected onto a subspace (but noise is not projected!) is sufficient to recover original performance.

## 5 Related Work

DP-ERM and DP-SCO are arguably the most well-studied areas of differential privacy [CMS11, KST12, BST14, SCS13, WYX17, FTS17, BFTT19, MRTZ17, ZZMW17, WLK+17, FKT20, INS+19, BFGT20, STT20, LL21, AFKT21, BGN21, GTU22, GLL22]. Tight dependence on the number of model parameters and the number of samples is known for both DP-ERM [BST14] and DP-SCO [BFTT19]. In particular, for the error on general convex losses, an explicit polynomial dependence on the number of optimization variables is necessary. However, it is shown that if gradients lie in a fixed low-rank subspace $M$, the dependence on dimension $d$ can be replaced by rank$(M)$ which can be significantly smaller [JT14, STT20]. We extend this line of work to show that under a weaker assumption (restricted Lipschitz continuity with decaying coefficients) one can obtain analogous error guarantees that are independent of $d$, but do not require the gradients of the loss to strictly lie in any fixed low-rank subspace $M$. As a consequence, our results provide a plausible explanation for the empirical observation that dense fine-tuning can be effective and that fine-tuning a larger model under DP can generally be more advantageous in terms of utility than fine-tuning a smaller model [LTLH21, YNB+21]. A concurrent work shows that the standard dimension dependence of DP-SGD can be replaced by a dependence on the trace of the Hessian assuming the latter quantity is uniformly bounded [MMZ22].

A complementary line of work designed variants of DP-SGD that either explicitly or implicitly control the subspace in which gradients are allowed to reside [AGM+21, LVS+21, ADF+21, KDRT21, YZCL21b]. They demonstrated improved dependence of the error on the dimension if the true gradients lie in a "near" low-rank subspace. Our results are incomparable to this line of work because of two reasons: (i) Our algorithm is vanilla DP-SGD and does not track the gradient subspace either explicitly or implicitly, and hence does not change the optimization landscape. Our improved dependence on dimensions is an artifact of the analysis. (ii) Our analytical results do not need the existence of any public data to obtain tighter dependence on dimensions. All prior works mentioned above need the existence of public data to demonstrate any improvement.

On the empirical front, past works have observed that for image classification tasks, gradients of ConvNets converge to a small subspace spanned by the top directions of the Hessian. In addition, this span remains stable for long periods of time during training [GARD18]. While insightful, this line of work does not look at language model fine-tuning. Another line of work measures for language model fine-tuning the *intrinsic dimension*—the minimum dimension such that optimizing in a randomly sampled subspace of such dimension approximately recovers the original performance [LFLY18, AZG20]. We note that a small intrinsic dimension likely suggests that gradients are approximately low rank. Yet, this statement should not be interpreted as a strict implication, since the notion of

intrinsic dimension is at best vaguely defined (e.g., there's no explicit failure probability threshold over the randomly sampled subspace in the original statement), and the definition involves not a fixed subspace but rather a randomly sampled one.

# 6   Conclusion

We made an attempt to reconcile two seemingly conflicting results: (i) in private convex optimization, errors are predicted to scale proportionally with the dimension of the learning problem; while (ii) in empirical works on large-scale private fine-tuning through DP-SGD, privacy-utility trade-offs become better with increasing model size. We introduced the notion of restricted Lipschitz continuity, with which we gave refined analyses of DP-SGD for DP-ERM and DP-SCO. When the magnitudes of gradients projected onto diminishing subspaces decay rapidly, our analysis showed that excess empirical and population losses of DP-SGD are independent of the model dimension. Through preliminary experiments, we gave empirical evidence that gradients of large pretrained language models obtained through fine-tuning mostly lie in the subspace spanned by a few principal components. Our theoretical and empirical results together give a possible explanation for recent successes in large-scale differentially private fine-tuning.

Given our improved upper bounds on the excess empirical and population risks for differentially private convex learning, it is instructive to ask if such bounds are tight in the mini-max sense. We leave answering this inquiry to future work. In addition, while we have presented encouraging empirical evidence that fine-tuning gradients mostly lie in a small subspace, more work is required to study the robustness of this phenomenon with respect to the model class and fine-tuning problem. Overall, we hope that our work leads to more research on understanding conditions under which DP learning does not degrade with increasing problem size, and more generally, how theory can inform and explain the practical successes of differentially private deep learning.

# Acknowledgements

XL is supported by a Stanford Graduate Fellowship.

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
