# Appendix

## A  Proof of Theorem 3.3

Before bounding the utility of DP-SGD, we first bound $x^{(t)} - x^{(0)}$ in expectation. For simplicity, we write $f_j(\cdot)$ for $f(\cdot; s_j)$ and $F(\cdot)$ for $F(\cdot; \mathcal{D})$ when there is no ambiguity.

**Lemma A.1.** *Suppose Assumption 3.2 holds. Let $Q$ be any orthogonal projection matrix with rank $r$ and suppose that $\|Q\nabla f(x; s)\|_2 \leq G_Q$ for all $x \in \mathbb{R}^d$ and $s \in \mathcal{S}$. If we set $\eta \leq \frac{1}{2\alpha}$ in DP-SGD, then for all $t > 0$, we have*

$$\mathbb{E}\|Q(x^{(t)} - x^{(0)})\|_2^2 \leq \frac{4G_Q^2}{\alpha^2} + \frac{2\eta G_0^2}{\alpha}(1 + r\sigma^2).$$

*Proof of Lemma A.1.*  By the assumption, we know $\|Q\nabla F(x)\|_2 \leq G_Q$ and $\|\nabla F(x)\|_2 \leq G_0$. Let $z^{(t)} = x^{(t)} - x^{(0)}$. Note that

$$
\begin{aligned}
z^{(t+1)} &= x^{(t+1)} - x^{(0)} \\
&= x^{(t)} - \eta\Big(\nabla f_j(x^{(t)}) + \alpha(x^{(t)} - x^{(0)}) + G_0 \cdot \zeta\Big) - x^{(0)} \\
&= (1 - \alpha\eta)z^{(t)} - \eta(\nabla f_j(x^{(t)}) + G_0 \cdot \zeta),
\end{aligned}
$$

where $\zeta \sim \mathcal{N}(0, \sigma^2 I_d)$ is the isotropic Gaussian noise drawn in $(t+1)$th step. For simplicity, we use $\tilde{\nabla} f_j(x^{(t)})$ to denote the noisy subgradient $\nabla f_j(x^{(t)}) + G_0 \cdot \zeta$. Hence, we have

$$\|Qz^{(t+1)}\|_2^2 = (1 - \alpha\eta)^2\|Qz^{(t)}\|_2^2 - 2\eta(1 - \alpha\eta)(Qz^{(t)})^\top Q\tilde{\nabla} f_j(x^{(t)}) + \eta^2\|Q\tilde{\nabla} f_j(x^{(t)})\|_2^2.$$

Taking expectation over the random sample $f_j$ and random Gaussian noise $\zeta$, we have

$$
\begin{aligned}
\mathbb{E}\|Qz^{(t+1)}\|_2^2 =&(1 - \alpha\eta)^2\,\mathbb{E}\|Qz^{(t)}\|_2^2 - 2\eta(1 - \alpha\eta)\cdot\mathbb{E}\Big((Qz^{(t)})^\top(Q\nabla F(x^{(t)}))\Big) \\
&+ \eta^2\,\mathbb{E}\|Q\tilde{\nabla} f_j(x^{(t)})\|_2^2 \\
\leq&(1 - \alpha\eta)\,\mathbb{E}[\|Qz^{(t)}\|_2^2] + 2\eta G_Q\cdot\mathbb{E}[\|Qz^{(t)}\|_2] + \eta^2 G_0^2(1 + r\sigma^2),
\end{aligned}
$$

where we used the fact that $\zeta$ has zero mean, $\|\nabla f_j(x^{(t)})\|_2 \leq G_0$, $\|Q\nabla F(x^{(t)})\|_2 \leq G_Q$ and $\eta \leq \frac{1}{2\alpha}$. Further simplifying and taking expectation over all iterations, we have

$$
\begin{aligned}
\mathbb{E}\|Qz^{(t+1)}\|_2^2 &\leq (1 - \alpha\eta)\,\mathbb{E}\|Qz^{(t)}\|_2^2 + 2\eta\big(\frac{\alpha}{4}\,\mathbb{E}\|Qz^{(t)}\|_2^2 + \frac{1}{\alpha}G_Q^2\big) + \eta^2 G_0^2(1 + r\sigma^2) \\
&\leq (1 - \frac{\alpha\eta}{2})\,\mathbb{E}\|Qz^{(t)}\|_2^2 + \frac{2\eta}{\alpha}G_Q^2 + \eta^2 G_0^2(1 + r\sigma^2). \quad\quad (5)
\end{aligned}
$$

Using that $z^{(0)} = 0$ we know $\mathbb{E}\|Qz^{(0)}\|_2^2 = 0$. Solving the recursion (Equation (5)) gives

$$\mathbb{E}\|Qz^{(t)}\|_2^2 \leq \frac{2}{\alpha\eta}\big(\frac{2\eta}{\alpha}G_Q^2 + \eta^2 G_0^2(1 + r\sigma^2)\big)$$

for all $t$. This proves the result.  $\square$

Now, we are ready to bound the utility. The proof builds off the standard mirror descent proof.

**Lemma A.2.** *Let $\delta \in (0, \frac{1}{2}]$, and $\varepsilon \in (0, 10]$. Under Assumption 3.2, let $x^{(0)}$ be the initial iterate and $x^* = \arg\min_x F(x)$. Suppose $\|x^{(0)} - x^*\|_2 \leq D$. For any positive integer $k \leq d$, setting $T = \Theta(n^2 + d\log^2 d)$, $\sigma = \Theta\left(\frac{\sqrt{T\log(1/\delta)}}{n\varepsilon}\right)$, $\eta = \sqrt{\frac{D^2}{T \cdot G_0^2 \cdot k\sigma^2}}$ and $\alpha = \frac{1}{D}\sqrt{\sum_{s=1}^S s^2 2^s G_{2^{s-1}k}^2}$, we have*

$$\mathbb{E}[F(\overline{x}) - F(x^*)] \lesssim \frac{G_0 D\sqrt{k\log(1/\delta)}}{\varepsilon n} + D\sqrt{\sum_{s=1}^S s^2 2^s G_{2^{s-1}k}^2},$$

where $S = \lfloor \log(d/k) \rfloor + 1$, $\overline{x}$ is the output of DP-SGD, and the expectation is under the randomness of DP-SGD.

Moreover, if $G_k \leq G_0 k^{-c}$ for each $k$ for some $c > 1/2$, and in addition $n > \varepsilon^{-1}\sqrt{\log(1/\delta)}$, picking the best $k \in [d]$ for the bound above gives

$$\mathbb{E}[F(\overline{x}; \mathcal{D}) - F(x^*; \mathcal{D})] \lesssim G_0 D \cdot \left( \frac{\sqrt{\log(1/\delta)}}{\varepsilon n} \right)^{2c/(1+2c)}.$$

*Proof of Lemma A.2.* Fix a positive integer $1 \leq k \leq d$. Our key idea is to split the whole space $\mathbb{R}^d$ into different subspaces. We define the following set of subspaces:

- $U_0 = \mathrm{range}(P_k)$.

- For $s = 1, 2, \ldots, \lfloor \log(d/k) \rfloor$, let $U_s \subseteq \mathrm{range}(P_{2^s k})$ be a subspace with maximal dimension such that $U_s \perp U_i$ for all $i = 0, \ldots, s-1$.

- For $S = \lfloor \log(d/k) \rfloor + 1$, let $U_S \subseteq \mathbb{R}^d$ be the subspace such that $\bigoplus_{i=0}^{S} U_i = \mathbb{R}^d$, and $U_S \perp U_i$ for all $i = 0, \ldots, S-1$.

Recall $P_i$ is the orthogonal projection matrix with rank $i$ that gives rise to $G_i$ in Assumption 3.2. In the above, we have assumed that the base of log is 2.

Let $Q_s$ be the orthogonal projection matrix that projects vectors onto the subspace $U_s$. Note that $\mathrm{rank}(Q_s) \leq 2^s k$ since $U_s \subseteq \mathrm{range}(P_{2^s k})$.

Moreover, it's clear that $U_s \perp \mathrm{range}(P_{2^{s-1}k})$ for all $s \in \{1, \ldots, S\}$. This is because by construction $\bigoplus_{i=0}^{s-1} U_i \supseteq \mathrm{range}(P_{2^{s-1}k})$, and that $U_s \perp \bigoplus_{i=0}^{s-1} U_i$.

Thus,

$$\|Q_s \nabla F(x)\|_2 = \|Q_s(I - P_{2^{s-1}k})\nabla F(x)\|_2 \leq \|Q_s\|_{\mathrm{op}} \|(I - P_{2^{s-1}k})\nabla F(x)\|_2 \leq G_{2^{s-1}k} \quad (6)$$

for all $x \in \mathbb{R}^d$ and all $s \in \{1, \ldots, S\}$.

Let $j \in [n]$ be the (uniformly random) index sampled in iteration $t$ of DP-SGD. By convexity of the individual loss $f_j$,

$$f_j(x^{(t)}) - f_j(x^*) \leq \nabla f_j(x^{(t)})^\top (x^{(t)} - x^*).$$

By construction, $\mathbb{R}^d$ is the orthogonal direct sum of the subspaces $\{U_j\}_{j=0}^{S}$, and thus any vector $v \in \mathbb{R}^d$ can be rewritten as the sum $\sum_{i=0}^{S} Q_i v$. We thus split the right hand side of the above as follows

$$f_j(x^{(t)}) - f_j(x^*) \leq \left( Q_0 \nabla f_j(x^{(t)}) + \sum_{s=1}^{S} Q_s \nabla f_j(x^{(t)}) \right)^\top (x^{(t)} - x^*). \quad (7)$$

We use different approaches to bound $(Q_0 \nabla f_j(x^{(t)}))^\top (x^{(t)} - x^*)$ and $(Q_s \nabla f_j(x^{(t)}))^\top (x^{(t)} - x^*)$ when $s \geq 1$, and we discuss them separately in the following.

**Term one:** Bound $(Q_0 \nabla f_j(x^{(t)}))^\top (x^{(t)} - x^*)$. Recall that

$$x^{(t+1)} = x^{(t)} - \eta \left( \nabla f_j(x^{(t)}) + \alpha(x^{(t)} - x^{(0)}) + G_0 \cdot \zeta \right)$$

for some Gaussian $\zeta \sim \mathcal{N}(0, \sigma^2 I_d)$. Hence, we have

$$(\nabla f_j(x^{(t)}))^\top Q_0 (x^{(t)} - x^*)$$
$$= \left( \frac{1}{\eta}(x^{(t)} - x^{(t+1)}) - \alpha(x^{(t)} - x^{(0)}) - G_0 \cdot \zeta \right)^\top Q_0(x^{(t)} - x^*)$$
$$= \left( \frac{1}{\eta}Q_0(x^{(t)} - x^{(t+1)}) \right)^\top Q_0(x^{(t)} - x^*) - \left( \alpha(x^{(t)} - x^{(0)}) + G_0 \cdot \zeta \right)^\top Q_0(x^{(t)} - x^*)$$

$$=\frac{1}{2\eta}\left(\|Q_0(x^{(t)}-x^*)\|_2^2-\|Q_0(x^{(t+1)}-x^*)\|_2^2+\|Q_0(x^{(t)}-x^{(t+1)})\|_2^2\right)$$

$$-\left(\alpha(x^{(t)}-x^{(0)})+G_0\cdot\zeta\right)^\top Q_0(x^{(t)}-x^*),\tag{8}$$

where we used the fact that $Q_0^2 v = Q_0 v$ for any $v \in \mathbb{R}^d$ (since $Q_0$ is a projection matrix), and the last equality follows from that

$$2(Q_0(x^{(t)}-x^{(t+1)}))^\top Q_0(x^{(t)}-x^*)$$
$$=\|Q_0(x^{(t)}-x^*)\|_2^2-\|Q_0(x^{(t+1)}-x^*)\|_2^2+\|Q_0(x^{(t)}-x^{(t+1)})\|_2^2.$$

Taking expectation on $\zeta$ over both sides of Equation (8) and making use of the fact that $\zeta$ has mean 0, we have

$$\underset{\zeta}{\mathbb{E}}(Q_0\nabla f_j(x^{(t)}))^\top(x^{(t)}-x^*)$$

$$=\frac{1}{2\eta}\left(\underset{\zeta}{\mathbb{E}}\|Q_0(x^{(t)}-x^*)\|_2^2-\underset{\zeta}{\mathbb{E}}\|Q_0(x^{(t+1)}-x^*)\|_2^2+\underset{\zeta}{\mathbb{E}}\|Q_0(x^{(t)}-x^{(t+1)})\|_2^2\right)$$

$$-\alpha\underset{\zeta}{\mathbb{E}}\left((x^{(t)}-x^{(0)})^\top Q_0(x^{(t)}-x^*)\right).$$

Recalling the definition of $Q_0$ and that $Q_0$ has rank at most $k$, one has

$$\underset{\zeta}{\mathbb{E}}\|Q_0(x^{(t)}-x^{(t+1)})\|_2^2=\eta^2\underset{\zeta}{\mathbb{E}}\|Q_0\left(\nabla f_j(x^{(t)})+\alpha(x^{(t)}-x^{(0)})+G_0\cdot\zeta\right)\|_2^2$$

$$=\eta^2\underset{\zeta}{\mathbb{E}}\|Q_0\left(\nabla f_j(x^{(t)})+\alpha(x^{(t)}-x^{(0)})\right)\|_2^2+\eta^2 G_0^2 k\sigma^2$$

$$\leq 2\eta^2 G_0^2(1+k\sigma^2)+2\eta^2\alpha^2\underset{\zeta}{\mathbb{E}}\|Q_0(x^{(t)}-x^{(0)})\|_2^2.$$

Moreover, one has

$$-\alpha(x^{(t)}-x^{(0)})^\top Q_0(x^{(t)}-x^*)$$

$$=-\alpha(x^{(t)}-x^{(0)})^\top Q_0(x^{(t)}-x^{(0)})-\alpha(x^{(t)}-x^{(0)})^\top Q_0(x^{(0)}-x^*)$$

$$\leq-\frac{\alpha}{2}\|Q_0(x^{(t)}-x^{(0)})\|_2^2+\frac{\alpha}{2}\|Q_0(x^{(0)}-x^*)\|_2^2.$$

Hence, we have

$$\underset{\zeta}{\mathbb{E}}(Q_0\nabla f_j(x^{(t)}))^\top(x^{(t)}-x^*)$$

$$\leq\frac{1}{2\eta}\left(\underset{\zeta}{\mathbb{E}}\|Q_0(x^{(t)}-x^*)\|_2^2-\underset{\zeta}{\mathbb{E}}\|Q_0(x^{(t+1)}-x^*)\|_2^2\right)+\eta G_0^2(1+k\sigma^2)$$

$$+\eta\alpha^2\underset{\zeta}{\mathbb{E}}\|Q_0(x^{(t)}-x^{(0)})\|_2^2-\frac{\alpha}{2}\underset{\zeta}{\mathbb{E}}\|Q_0(x^{(t)}-x^{(0)})\|_2^2+\frac{\alpha}{2}\underset{\zeta}{\mathbb{E}}\|Q_0(x^{(0)}-x^*)\|_2^2$$

$$\leq\frac{1}{2\eta}\left(\underset{\zeta}{\mathbb{E}}\|Q_0(x^{(t)}-x^*)\|_2^2-\underset{\zeta}{\mathbb{E}}\|Q_0(x^{(t+1)}-x^*)\|_2^2\right)+\eta G_0^2(1+k\sigma^2)+\frac{\alpha}{2}\underset{\zeta}{\mathbb{E}}\|Q_0(x^{(0)}-x^*)\|_2^2,$$

where we used $\eta\leq\frac{1}{2\alpha}$ at the end.

**Term two:** Bound $(Q_s\nabla f_j(x^{(t)}))^\top(x^{(t)}-x^*)$. We bound the objective above for each $s$ separately. By taking expectation over the random $f_j$, we have

$$\underset{f_j}{\mathbb{E}}(Q_s\nabla f_j(x^{(t)}))^\top(x^{(t)}-x^*)=(Q_s\nabla F(x^{(t)}))^\top(x^{(t)}-x^*)$$

$$\leq\|Q_s\nabla F(x^{(t)})\|_2\cdot\|Q_s(x^{(t)}-x^*)\|_2$$

$$\leq\frac{1}{\alpha_s}\|Q_s\nabla F(x^{(t)})\|_2^2+\frac{\alpha_s}{4}\|Q_s(x^{(t)}-x^*)\|_2^2$$

$$\leq\frac{G_{2^{s-1}k}^2}{\alpha_s}+\frac{\alpha_s}{2}\|Q_s(x^{(t)}-x^{(0)})\|_2^2+\frac{\alpha_s}{2}\|Q_s(x^{(0)}-x^*)\|_2^2,$$

where we chose $\alpha_s = \alpha s^{-2} 2^{-s}$ and used the bound (6) and Young's inequality at the end.

**Combination:** Combining both terms into (7) and taking expectation over all randomness, we have

$$\mathbb{E}[F(x^{(t)}) - F(x^*)]$$

$$\leq \frac{1}{2\eta}(\mathbb{E}\|Q_0(x^{(t)} - x^*)\|_2^2 - \mathbb{E}\|Q_0(x^{(t+1)} - x^*)\|_2^2) + \eta G_0^2(1 + k\sigma^2) + \frac{\alpha}{2}\mathbb{E}\|Q_0(x^{(0)} - x^*)\|_2^2$$

$$+ \sum_{s=1}^{S} \frac{G_{2^{s-1}k}^2}{\alpha_s} + \frac{1}{2}\sum_{s=1}^{S}\alpha_s \mathbb{E}\|Q_s(x^{(t)} - x^{(0)})\|_2^2 + \frac{1}{2}\sum_{s=1}^{S}\alpha_s \mathbb{E}\|Q_s(x^{(0)} - x^*)\|_2^2$$

$$\leq \frac{1}{2\eta}(\mathbb{E}\|Q_0(x^{(t)} - x^*)\|_2^2 - \mathbb{E}\|Q_0(x^{(t+1)} - x^*)\|_2^2) + \eta G_0^2(1 + k\sigma^2) + \frac{\alpha}{2}\|x^{(0)} - x^*\|_2^2$$

$$+ \sum_{s=1}^{S} \frac{G_{2^{s-1}k}^2}{\alpha_s} + \frac{1}{2}\sum_{s=1}^{S}\alpha_s \mathbb{E}\|Q_s(x^{(t)} - x^{(0)})\|_2^2.$$

Recall $\alpha \cdot \eta \leq 1/2$. Under the other assumptions, by Lemma A.1, one can show

$$\mathbb{E}\|Q_s(x^{(t)} - x^{(0)})\|^2 \leq \frac{4G_{2^{s-1}k}^2}{\alpha^2} + \frac{2\eta G_0^2}{\alpha}(1 + 2^s k\sigma^2)$$

$$\leq \frac{4G_{2^{s-1}k}^2}{\alpha_s^2} + \frac{2\eta G_0^2}{\alpha_s s^2}(1 + k\sigma^2).$$

Using $\sum_{s=1}^{\infty} s^{-2} \leq 2$, we have

$$\mathbb{E}\,F(x^{(t)}) - \mathbb{E}\,F(x^*)$$

$$\leq \frac{1}{2\eta}(\mathbb{E}\|Q_0(x^{(t)} - x^*)\|_2^2 - \mathbb{E}\|Q_0(x^{(t+1)} - x^*)\|_2^2) + \frac{\alpha}{2}\|x^{(0)} - x^*\|^2$$

$$+ \eta G_0^2(1 + k\sigma^2) + 3\sum_{s=1}^{S} \frac{G_{2^{s-1}k}^2}{\alpha_s} + 2\eta G_0^2(1 + k\sigma^2)$$

$$\leq \frac{1}{2\eta}(\mathbb{E}\|Q_0(x^{(t)} - x^*)\|_2^2 - \mathbb{E}\|Q_0(x^{(t+1)} - x^*)\|_2^2) + \frac{\alpha}{2}\|x^{(0)} - x^*\|_2^2$$

$$+ 3\eta G_0^2(1 + k\sigma^2) + \frac{3}{\alpha}\sum_{s=1}^{S} s^2 2^s G_{2^{s-1}k}^2.$$

Summing up over $t = 1, 2, \cdots, T$, by the assumption that $\|x^{(0)} - x^*\|_2 \leq D$ and convexity of the function, we have

$$\mathbb{E}[F(\overline{x}) - F(x^*)] \leq \frac{D^2}{2\eta T} + 3\eta G_0^2(1 + k\sigma^2) + \frac{\alpha}{2}D^2 + \frac{3}{\alpha}\sum_{s=1}^{S} s^2 2^s G_{2^{s-1}k}^2. \tag{9}$$

Set the parameters $T = c_1(n^2 + d\log^2 d)$, $\sigma = \frac{c_2\sqrt{T\log(1/\delta)}}{n\varepsilon}$, $\eta = \sqrt{\frac{D^2}{T \cdot G_0^2 \cdot k\sigma^2}}$ and $\alpha = \frac{1}{D}\sqrt{\sum_{s=1}^{S} s^2 2^s G_{2^{s-1}k}^2}$ for some large constants $c_1, c_2$. Note that this choice of parameters satisfies the condition of

$$\eta \cdot \alpha = \sqrt{\frac{D^2}{T \cdot G_0^2 \cdot k\sigma^2}} \cdot \frac{\sqrt{\sum_{s=1}^{S} s^2 2^s G_{2^{s-1}k}^2}}{D}$$

$$\leq \sqrt{\frac{G_0^2(2d)\log^3(2d)}{T \cdot G_0^2 \cdot k\sigma^2}} = \frac{n\varepsilon}{c_2 T}\sqrt{\frac{(2d)\log^3(2d)}{k \cdot \log(1/\delta)}}$$

$$\leq \frac{n\varepsilon\sqrt{(2d)\log^3(2d)}}{c_2 T} \leq \frac{1}{2},$$

where we used the fact that $G_k \leq G_0$, $s \leq S \leq \log(2d)$, $T \geq n^2 + d\log^2 d$, and $c_2$ is large enough. Using the parameters we pick, we have

$$\mathbb{E}[F(\overline{x}) - F(x^*)] \lesssim \frac{G_0 D \sqrt{k\log(1/\delta)}}{\varepsilon n} + D\sqrt{\sum_{s=1}^{S} s^2 2^s G_{2^{s-1}k}^2}$$

Moreover, assuming $G_k \leq G_0 k^{-c}$ for some $c > 1/2$, we have $\sqrt{\sum_s s^2 2^s G_{2^{s-1}k}^2} \lesssim G_0/k^c$. Hence,

$$\mathbb{E}[F(\overline{x}) - F(x^*)] \lesssim \frac{G_0 D \sqrt{k\log(1/\delta)}}{\varepsilon n} + \frac{G_0 D}{k^c}.$$

Since the above bound holds for all $k \in \{1, \dots, d\}$, we may optimize it with respect to $k$. Recall by assumption that $n \geq \varepsilon^{-1}\sqrt{\log(1/\delta)}$. Letting

$$k = \min\left\{ d, \left\lceil \left( \frac{\varepsilon n}{\sqrt{\log(1/\delta)}} \right)^{\frac{2}{1+2c}} \right\rceil \right\}$$

yields the bound

$$\mathbb{E}[F(\overline{x}; \mathcal{D}) - F(x^*; \mathcal{D})] \lesssim G_0 D \cdot \left( \frac{\sqrt{\log(1/\delta)}}{\varepsilon n} \right)^{2c/(1+2c)}.$$

$\square$

Combining the privacy guarantee in Lemma 2.4 and Lemma A.2 directly results in Theorem 3.3.

## B  Proof of Theorem 3.5

We study the generalization error of DP-SGD. Similar to previous works, we make use of the stability of DP-SGD to bound its generalization error. The bound on the excess population loss follows from combining bounds on the excess empirical loss and the generalization error. Before stating the proof, we first recall two results in the literature.

**Lemma B.1** ([BE02, Lemma 7]). *Given a learning algorithm $\mathcal{A}$, suppose the dataset $\mathcal{D} = \{s_1, \cdots, s_n\}$ is made up of $n$ i.i.d. samples drawn from the underlying distribution $\mathcal{P}$, and we replace one random sample in $\mathcal{D}$ with a freshly sampled $s' \sim \mathcal{P}$ and get a new neighboring dataset $\mathcal{D}'$. One has*

$$\mathbb{E}_{\mathcal{D},\mathcal{A}}[F(\mathcal{A}(\mathcal{D}); \mathcal{P}) - F(\mathcal{A}(\mathcal{D}); \mathcal{D})] = \mathbb{E}_{\mathcal{D}, s'\sim\mathcal{P},\mathcal{A}}[f(\mathcal{A}(\mathcal{D}); s') - f(\mathcal{A}(\mathcal{D}'); s')],$$

*where $\mathcal{A}(\mathcal{D})$ is the output of $\mathcal{A}$ with input $\mathcal{D}$.*

**Lemma B.2** ([BFGT20, Theorem 3.3]). *Suppose Assumption 3.2 holds, running DP-SGD with step size $\eta$ on any two neighboring datasets $\mathcal{D}$ and $\mathcal{D}'$ for $T$ steps satisfies that*

$$\mathbb{E}\left[\|\overline{x} - \overline{x}'\|_2\right] \leq 4G_0\eta\left(\frac{T}{n} + \sqrt{T}\right),$$

*where $\overline{x}$ and $\overline{x}'$ are the outputs of DP-SGD with datasets $\mathcal{D}$ and $\mathcal{D}'$, respectively.*

*Proof of Theorem 3.5.* Let $\overline{x}$ and $\overline{x}'$ be the outputs of DP-SGD when applied to the datasets $\mathcal{D}$ and $\mathcal{D}'$, respectively. $\mathcal{D}'$ is a neighbor of $\mathcal{D}$ with one example replaced by $s' \sim \mathcal{P}$ that is independently sampled. Combining Lemma B.1 and Lemma B.2 yields

$$\mathbb{E}[F(\overline{x}; \mathcal{P}) - F(\overline{x}; \mathcal{D})] = \mathbb{E}[f(\overline{x}; s') - f(\overline{x}'; s')]$$
$$\leq \mathbb{E}[G_0 \|\overline{x} - \overline{x}'\|_2]$$

$$\leq 4G_0^2\eta\left(\frac{T}{n} + \sqrt{T}\right).$$

Similar to the DP-ERM case, by setting $T = c_1(n^2 + d\log^2 d)$, $\sigma = \frac{c_2\sqrt{T\log(1/\delta)}}{n\varepsilon}$, $\eta = \sqrt{\frac{D^2}{T\cdot G_0^2(T/n+k\sigma^2)}}$ and $\alpha = \frac{1}{D}\sqrt{\sum_{s=1}^S s^2 2^s G_{2^{s-1}k}^2}$ for some large positive constants $c_1$ and $c_2$, we conclude that $\eta \cdot \alpha \leq 1/2$. Hence, Equation (9) shows that, for any fixed dataset $\mathcal{D}$ and any $x^*$ such that $\left\|x^{(0)} - x^*\right\|_2 \leq D$, we have

$$\mathbb{E}[F(\overline{x}; \mathcal{D}) - F(x^*; \mathcal{D})] \leq \frac{D^2}{2\eta T} + 3\eta G_0^2(1 + k\sigma^2) + \frac{\alpha}{2}D^2 + \frac{3}{\alpha}\sum_{s=1}^S s^2 2^s G_{2^{s-1}k}^2.$$

We can therefore rewrite the population loss as follows

$$\begin{aligned}
&\mathbb{E}[F(\overline{x}; \mathcal{P}) - F(x^*; \mathcal{P})] \\
&= \mathbb{E}[F(\overline{x}; \mathcal{P}) - F(\overline{x}; \mathcal{D})] + \mathbb{E}_{\mathcal{D}}[F(\overline{x}; \mathcal{D}) - F(x^*; \mathcal{D})] \\
&\leq 4G_0^2\eta\left(\frac{T}{n} + \sqrt{T}\right) + \frac{D^2}{2\eta T} + 3\eta G_0^2(1 + k\sigma^2) + \frac{\alpha}{2}D^2 + \frac{3}{\alpha}\sum_{s=1}^S s^2 2^s G_{2^{s-1}k}^2.
\end{aligned}$$

Substituting in the values for parameters $T$, $\sigma$, $\eta$, and $\alpha$ yields

$$\mathbb{E}[F(\overline{x}; \mathcal{P}) - F(x^*; \mathcal{P})] \lesssim \frac{G_0 D}{\sqrt{n}} + \frac{G_0 D\sqrt{k\log(1/\delta)}}{\varepsilon n} + D\sqrt{\sum_{s=1}^\infty s^2 2^s G_{2^{s-1}k}^2}$$

for all positive integers $k \leq d$.

Similarly, if we have $G_k \leq G_0 k^{-c}$ for some $c > 1/2$, and in addition $n > \varepsilon^{-1}\log(1/\delta)$, it immediately follows that

$$\mathbb{E}[F(\overline{x}; \mathcal{P}) - \min_x F(x; \mathcal{P})] \lesssim \frac{G_0 D}{\sqrt{n}} + G_0 D \cdot \left(\frac{\sqrt{\log(1/\delta)}}{\varepsilon n}\right)^{2c/(1+2c)}.$$

This completes the proof. $\qquad\square$

## C Protocol for Synthetic Example Experiments in Section 4.1

We detail the construction of the synthetic example in Section 4.1. The training and test sets of this example both have $n_{\text{train}} = n_{\text{test}} = 10000$ instances. Each instance $x_i$ is sampled from a distribution where the first $d_{\min} = 10$ coordinates are all multi-variate normal distributions with mean and standard deviation both being 1. All remaining coordinates are constantly 0. This ensures the optimal non-private training losses for problems of different dimensions are the same.

## D Protocol and Additional Fine-Tuning Experiments for Section 4.2

### D.1 Experimental Protocol

For experiments in Section 4.2, we fine-tuned the DistilRoBERTa model with $(8, 1/n^{1.1})$-DP on the SST-2 training set with $n \geq 60000$ examples and measured performance on the companion dev set. We used the exact set of hyperparameters presented in [LTLH21] for this task. We repeated our re-training experiments over five independent random seeds. Fine-tuning for 3 epochs on this task takes 10 minutes on an A100-powered machine with sufficient RAM and CPU cores.

Our spectral analysis relies on running the orthogonal iteration algorithm on the set of collected gradients along the private fine-tuning trajectory [Dem97].[4] Unlike other numerical algorithms for estimating eigenvalues of a matrix, orthogonal iteration provably produces the set of eigenvalues that are largest in absolute value (and corresponding eigenvectors, if the underlying matrix is normal) in the limit.[5] Recall that we needed the top eigenvectors for projection in our re-training experiment.[6] By default, we run orthogonal iteration for ten iterations. We show in the following that our results are insensitivity to the number of iterations used in the orthogonal iteration algorithm. Each orthogonal iteration takes 10 minutes on an A100-powered machine with $r = 4000$ gradient samples and $k = 1000$ principal components for the DistilRoBERTa experiments.

### D.2 Robustness of Results

**Robustness to the number of orthogonal iterations.** The orthogonal iteration algorithm is a generalization of the power method that simultaneously produces estimates of multiple eigenvalues. Its convergence is known to be sensitivity to the gap between successive eigenvalues. The algorithm converges slowly if consecutive eigenvalues (with the largest absolute values) are close in absolute value. To confirm that our results aren't sensitivity to the choice of the number of iterations, we visualize the top 500 eigenvalues for the orthogonal iteration algorithm is run for different number of updates. Figure 3 shows that the linear fit to the top 500 eigenvalues remains stable across different number of orthogonal iterations $T$. Notably, $T = 10$ produces similar results as $T = 100$. These results were obtained with $r = 4000$ gradients.

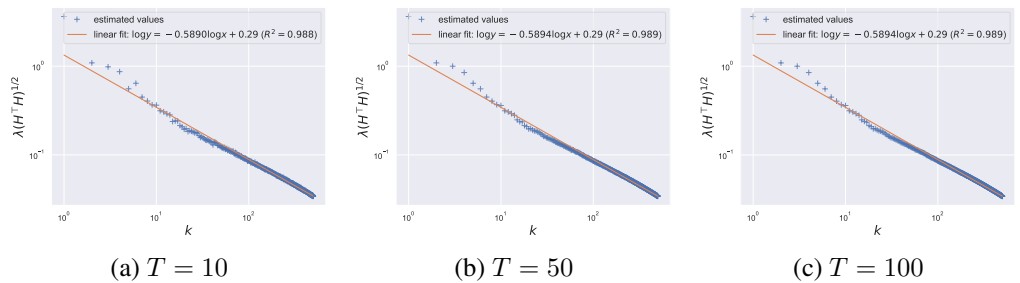

|  |  |  |
|---|---|---|
| (a) $T = 10$ | (b) $T = 50$ | (c) $T = 100$ |

Figure 3: The eigenspectrum remains stable with different number of iterations $T$.

---

[4]By gradients, we always mean the average of clipped per-example gradients — before adding noise — in this section.

[5]The orthogonal iteration algorithm is also known as simultaneous iteration or subspace iteration.

[6]Note that one commonly used algorithm in the neural net spectral analysis literature — Lanczos iteration [GKX19] — does not guarantee that the top eigenvalues are produced, even though its spectral estimates are frequently deemed accurate [GWG19].

**Robustness to the number of gradient samples.** We further ran experiments with different numbers of gradient samples $r$ collected along the fine-tuning trajectory, and plot the top 500 eigenvalues. Figure 4 shows that while the slope and intercept of the fitted line in log-log space changes, the change is moderate. Notably, the decaying trend of the top eigenvalues remains stable.

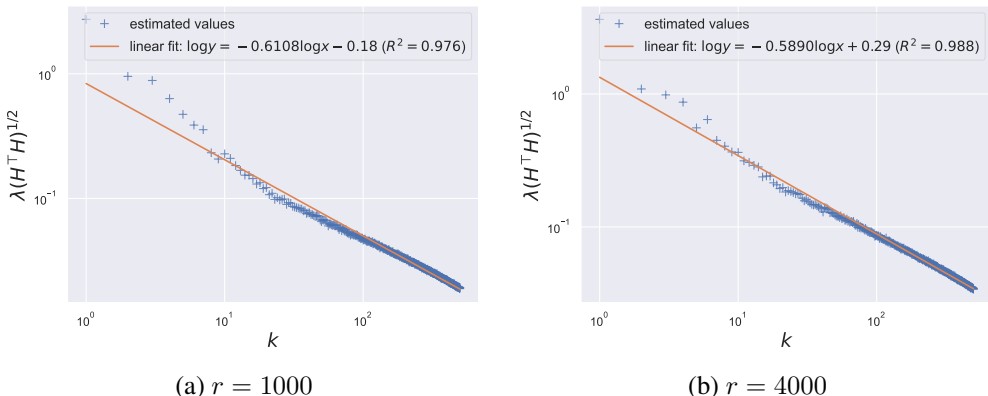

(a) $r = 1000$                          (b) $r = 4000$

Figure 4: The fast decaying trend of the eigenspectrum remains stable with different number of samples $r$ used for the spectral analysis.

**Robustness to the gradient sampling strategy.** We observe that gradients at the beginning of fine-tuning tend to be larger in magnitude than gradients collected later on along the optimization trajectory. To eliminate the potential confounder that the top principal components are solely formed by the initial few gradients evaluated during fine-tuning, we re-ran the spectral analysis experiment without the initial gradients. Specifically, we performed PCA for the gradients evaluated from step 300 to step 1300 during private fine-tuning, and compared the distribution of top eigenvalues returned from this setup to when we used the first 1000 gradients. Note the dev set accuracy converged to $\approx 90\%$ by step 200. Figure 5 shows that while the slope and intercept of linear fits are slightly different in the new setup compared to the old setup (when all gradients along the fine-tuning trajectory were used for PCA), that the eigenvalues follow a rapidly decaying trend remains true under both setups.

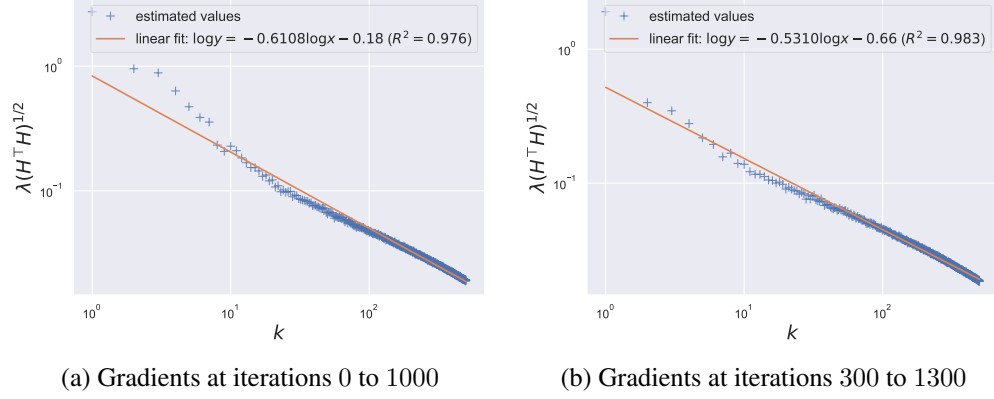

(a) Gradients at iterations 0 to 1000         (b) Gradients at iterations 300 to 1300

Figure 5: The rapidly decaying trend of the eigenspectrum remains stable with different sampling strategies.

**Robustness to model size.** In previous experiments, we empirically verified that gradients for fine-tuning DistilRoBERTa are near low rank. Here, we show that similar observations also hold for Roberta-base and Roberta-large when fine-tuning only the attention layers. The former setup has approximately 14 million trainable parameters, while the latter has approximately 50 million. Figures 6 and 7 illustrate these results.

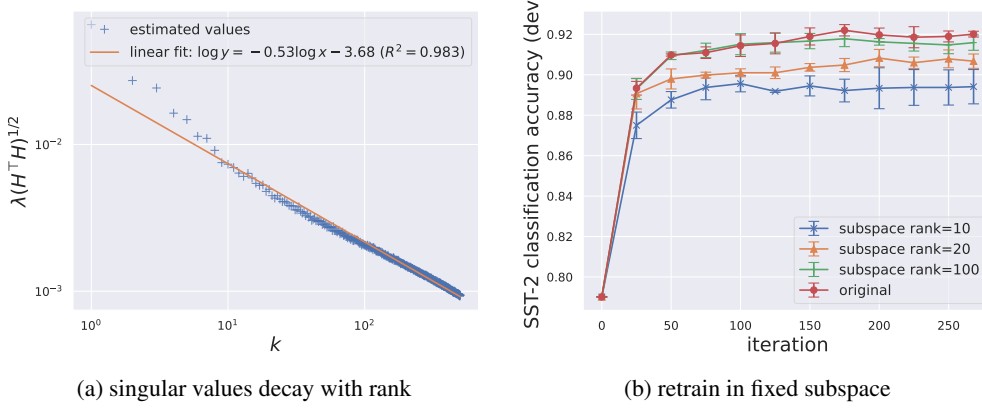

(a) singular values decay with rank

(b) retrain in fixed subspace

Figure 6: Experiments for Roberta-base.

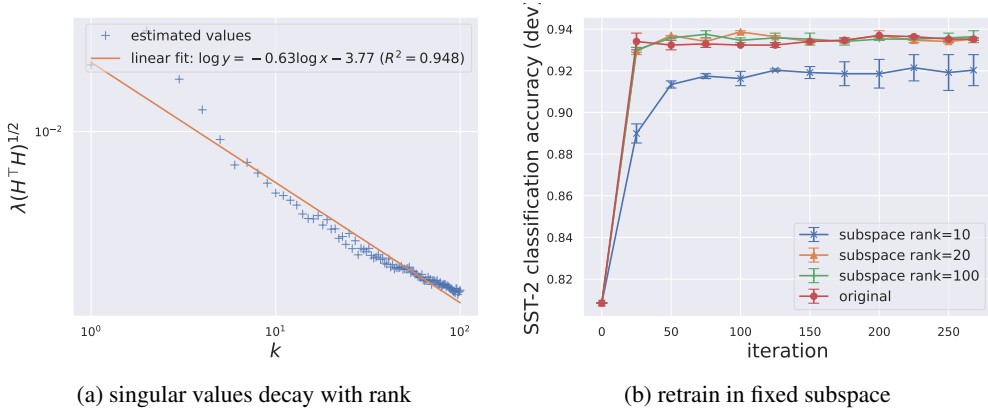

(a) singular values decay with rank

(b) retrain in fixed subspace

Figure 7: Experiments for Roberta-large.

**Non-robustness to fine-tuning strategy.** Recall our fine-tuning experiments for classification was based on the template-completion formulation detailed in [LTLH21]. As opposed to framing the task as integer label prediction, this formulation requires the model to predict one of $K$ candidate tokens to fill in a templated prompt for a $K$-way classification problem. While we have also performed the experiment where we re-train in the subspace of top principal components under the usual fine-tuning setup (stack a randomly initialized prediction head on top of the embedding of the [CLS] token), we found it difficult to recover the original fine-tuning performance when gradients are projected onto the top eigen-subspace with $d = 100$ dimensions. Retraining performance exhibited high variance and the final dev accuracy was bi-modal over random seeds with near guess accuracy ($\approx 50\%$) and original accuracy ($\approx 90\%$) being the two modes. We suspect this to be caused by the linear prediction head being randomly initialized.

### D.3 Additional Fine-Tuning Experiments with DistilGPT-2 for Generation Tasks

Experiments in Section 4.2 demonstrated that gradients from fine-tuning for classification are mostly controlled by a few principal components. In this section, we show that similar observations hold for fine-tuning on a text generation task. We follow the setup and hyperparameters in [LTLH21] for privately fine-tuning DistilGPT-2 on the E2E dataset [NDR17] under $(8, 1/n^{1.1})$-DP. We fine-tune all weight matrices in attention layers that produce the queries, values, and keys. This amounts to fine-tuning approximately 10.6 million parameters of a model with a total parameter count of more than 100 million. We again collected $r = 4000$ gradients evaluated during private fine-tuning, performed PCA, and conducted the eigenspectrum analysis. Figure 8 shows that the top eigenspectrum decays rapidly with a rate similar to what is observed in fine-tuning for the classification problem we studied.

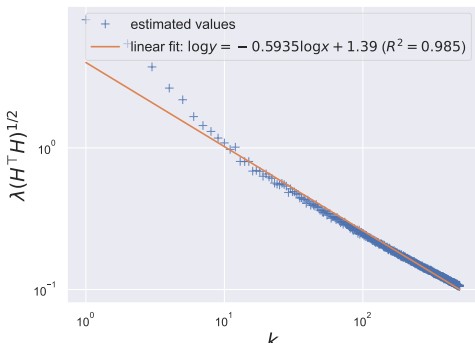

Figure 8: The eigenspectrum of gradients from fine-tuning for text generation rapidly decays.