# OpenReview forum: "When Does Differentially Private Learning Not Suffer in High Dimensions?"
_NeurIPS.cc/2022/Conference — NeurIPS 2022 Accept_

### Official Review · Reviewer_VSjU · 2022-07-10

**Rating:** 7
**Confidence:** 3
**Soundness:** 3 good
**Presentation:** 3 good
**Contribution:** 3 good

**Summary:**

Recent advances in natural language processing have witnessed the advantage of a large language model in private fine-tuning, which can achieve state-of-the-art performance even compared to non-private algorithms. Since the theoretical understanding of success is still lacking so far, this paper fills the gap to understand why the curse of dimension does not happen in the large-model fine-tuning. By using the novel notion of restricted Lipschitz-continuity, the authors proved a refined dimension-independent bound on the excess empirical error and excess population error by differentially-private stochastic gradient descent. Empirically, the authors show that the near-optimal models (e.g., pre-trained models) tend to produce gradients concentrated in a few principal components of the gradient subspace, which confirm the assumption of the restricted Lipschitz-continuity.

**Questions:**

1. The definition of the convex domain diameter D is not clearly stated. It was first used in Line 129 to state the prior bounds. In Theorem 3.3, D is defined to be the upper bound of the difference between x and x^*. It is confusing if the two D's denotes the same or related things. In the final excess error bounds, the D is used intensively both in prior arts and the paper's results. The authors may clarify the notation here.
2. In Line 267 of the Experiment section, why the DistillRoBERTa is selected? Do the results generalize to other large models? In the [LTLH21] which is the major empirical motivation for the work, the large RoBERTa was used. Some clarification may help the readers to understand the generality of the empirical studies. Since the restricted Lipschitz continuity is the major assumption of the paper, stronger empirical evidence is essential here.
3. How does the private fine-tuning of all attention parameters compared to more parameter-efficient fine-tuning? Since the dimension is a major concern for private learning in the scope of the paper, some advanced parameter-efficient fine-tuning should gain some advantage in the privacy-accuracy trade-off. By parameter-efficient fine-tuning, e.g., Prompt, the dimension of parameters to be privately trained could be effectively reduced. As a result, I believe that the authors may provide some empirical or theoretical results discussing the connection between the studied (almost) full fine-tuning and the parameter-efficient fine-tuning.

**Limitations:**

I did not find obvious negative social impacts of the work. The authors bring up the limitation of their work on the bound tightness. I believe the limitation is beyond the scope of the paper, which is reasonable to be left there.

**Strengths And Weaknesses:**

Strength
1. The paper asked an important and novel question: why the curse of dimension does not happen with private fine-tuning of pre-trained models? Prior empirical results have demonstrated the phenomenon, yet a theoretical understanding was not available. The paper fill the gap and clearly give an explanation of the source of the success.
2. Both the theoretical studies and empirical evaluations are insightful and sound and could inspire future works in the community of private learning.
3. The paper is well written and the experiments are well designed to support the theoretic results.

Weakness
1. My major concern is the experiment part. More large models should be evaluated to demonstrate the rationality of the restricted Lipschitz continuity assumption.

---

> ### Author Response · Authors · 2022-08-02
> **Response to reviewer VSjU part 1**
>
> We separately address the three questions raised by the reviewer.
>
> > The definition of the convex domain diameter D is not clearly stated. It was first used in Line 129 to state the prior bounds. In Theorem 3.3, D is defined to be the upper bound of the difference between x and x^*. It is confusing if the two D's denotes the same or related things. In the final excess error bounds, the D is used intensively both in prior arts and the paper's results.
>
> We thank the reviewer for pointing out this subtle issue. Indeed, the D on lines 128-129 means a different thing than the D defined in Theorem 3.3. On line 128-129, D refers to the diameter of the convex constraint set, whereas in Theorem 3.3, D is used as a shorthand for the distance between the initial iterate and the (global) optimum of the average loss. We will use two distinct symbols for the two concepts in the updated draft.
>
> > In Line 267 of the Experiment section, why the DistillRoBERTa is selected?
>
> We thank the reviewer for raising this question. Our experiments were constrained in scale mainly due to a need to perform spectrum analysis (simply stated, just PCA) of the gradients. Inspecting the distribution of eigenvalues provides us with insight for whether the gradients are ‘near’ low-rank.
>
> The orthogonal iteration algorithm that we adopted, in the limit, provably produces the top-k eigenvectors which have the largest eigenvalue in absolute value under mild regularity conditions. But this algorithm can be extremely memory-hungry if not implemented with special care.
>
> To provide the reviewer with an idea of the potential memory-intensiveness of the algorithm, we use our experimental setup in Section 4.2 as an example. The set of weights that we fine-tune for DistilRoberta has a total count of approximately 7 million. We store 4000 gradients for these parameters and run the orthogonal iteration algorithm to generate the top 1000 principal components. Storing the gradient matrix G (size 7 million x 4 k) together with the principal component matrix P (size 7 million x 1 k) in double precision alone consumes roughly 260 gigabytes of memory. Note such numerical algorithms can be potentially inaccurate when run in single precision due to the loss of precision. This substantial memory spending poses many challenges in terms of both RAM and VRAM.
>
> During the rebuttal period, we have improved our software infrastructure so that the spectrum computation can be performed in a highly distributed way. We have performed additional experiments for both Roberta-base and Roberta-large and will include these results in the updated supplementary material. The conclusion here is that things we observed for DistilRoberta also hold for the larger variants (i.e., decay of singular values and performance recovery through retraining in subspace).

---

> > ### Author Response · Authors · 2022-08-02
> > **Response to reviewer VSjU part 2**
> >
> > > How does the private fine-tuning of all attention parameters compared to more parameter-efficient fine-tuning? Since the dimension is a major concern for private learning in the scope of the paper, some advanced parameter-efficient fine-tuning should gain some advantage in the privacy-accuracy trade-off. By parameter-efficient fine-tuning, e.g., Prompt, the dimension of parameters to be privately trained could be effectively reduced. As a result, I believe that the authors may provide some empirical or theoretical results discussing the connection between the studied (almost) full fine-tuning and the parameter-efficient fine-tuning.
> >
> > We thank the reviewer for raising this question. The general question of whether parameter-efficient fine-tuning methods have a better privacy-utility trade-off has been extensively studied in [LTLH21, YNB+21]. In particular, it was shown in [LTLH21] Section 5 that methods which fine-tune fewer parameters might not perform better than a simple baseline which fine-tunes all parameters under the same privacy spending. Notably, a core underlying thesis of our current submission is also that the dimensionality of a private learning problem alone is not fully indicative of its performance. One vastly simplified example is a convex objective for high-dimensional inputs, where most directions have zero curvature. Perturbing along the zero curvature directions with noise does not degrade the loss value, even though the amount of noise injected through DP-optimization can be large.
> >
> > More specifically, the conclusion of the past works is that on simpler sentence classification problems in GLUE, parameter-efficient fine-tuning methods can have a small but concrete advantage (e.g., see Table 10 of [YNB+21]). Note this conclusion is subject to full fine-tuning potentially not being run with the best hyperparameters, as the authors of [LTLH21] did not tune hyperparameters on these classification tasks in order to avoid extra privacy leakage (the authors instead transferred hyperparameters tuned on another task). On the other hand, for the harder language generation tasks, parameter-efficient fine-tuning (prompt-based fine-tuning included) does not appear to possess a statistically significant advantage (see Table 2 of [LTLH21]).
> >
> > While it’s not our focus to study and compare different fine-tuning methods under differential privacy in this paper, we briefly comment on the reviewer’s question about comparing fine-tuning attention layers only to other fine-tuning methods. We experimented with fine-tuning only attention layers for Roberta-base under epsilon=8 without using the text infilling objective on SST-2 and get \~90% dev set accuracy. This is better than the numbers of full fine-tuning reported in [LTLH21] which is \~86%, and slightly worse than the numbers reported in [YNB+21] for lightweight fine-tuning methods such as RGP, Adapter, Compacter, and LoRA (\~91-92%). Note in all above setups, the noise multiplier used in DP-optimization is very similar; [YNB+21] report a smaller epsilon due to using a better privacy accounting technique.

---

> > > ### Comment · Reviewer_VSjU · 2022-08-05
> > > **Thank you for the replies**
> > >
> > > Thanks for the authors' replies, which address most of my concerns. I have updated my score.

---

> > > > ### Author Response · Authors · 2022-08-05
> > > > **Thank you**
> > > >
> > > > We thank the reviewer for their quick response and again for their time and feedback!

---

### Official Review · Reviewer_tNQa · 2022-07-11

**Rating:** 6
**Confidence:** 4
**Soundness:** 3 good
**Presentation:** 3 good
**Contribution:** 3 good

**Summary:**

This paper proposes an analysis that captures the behavior that DP-SGD’s utility performance does not deteriorate with the increasing dimension under a specific assumption. Such assumption generally states that if the gradient lies in a limited subspace of the ambient space, then the excess risk is then dimension-independent.
This paper gives some evidence for the fact that when fine-tuning a DL model, such an assumption holds. By such observation, the theoretical result explains some phenomena.



**Questions:**

See the limitations for details

**Limitations:**

This paper assumes that it is in the convex setting, however, even in the fine-turning stage, it is not guaranteed in the convex setting as is the case for deep learning model. Hence, such result has a limited implication.
Another important point is that, when doing fine-tuning, it is still not guaranteed to be near the local optimal point (we still don’t how to quantify “near” either). The experiment gives positive evidence, however, I think more experiment should be done to validate the hypothesis that fine-tuning starts from a near-optimal point.


**Strengths And Weaknesses:**

The error analysis technique is not new, and the key assumption proposed in this paper is a variant of the assumption that gradient lies in a limited subspace (which is not new either), however, from such an assumption to deriving the error bound seems like non-trivial. And such result has an implication.

---

> ### Author Response · Authors · 2022-08-02
> **Response to reviewer tNQa**
>
> We address two questions raised by the reviewer.
>
> > This paper assumes that it is in the convex setting, however, even in the fine-turning stage, it is not guaranteed in the convex setting as is the case for deep learning model. Hence, such result has a limited implication.
>
> We agree with the reviewer that there are limitations of using the analysis in convex settings to improve our understanding of optimization and generalization of non-convex deep learning models. We will further clarify this limitation in the main draft. However, we believe that our analyses still provide ample insight that can be leveraged to deepen our understanding of the near dimension-independent phenomena in differentially private deep learning. In particular, what remains relevant to both our theoretical analysis and empirical results is the near low-dimensional structure of gradients.
>
> We also note that there are many success stories in the deep learning literature which derive fine-grained quantitative insights from convex toy models to understand and predict key behaviours of optimization and generalization of deep neural networks [1, 2, 3, 4].
>
> > Another important point is that, when doing fine-tuning, it is still not guaranteed to be near the local optimal point (we still don’t how to quantify “near” either). The experiment gives positive evidence, however, I think more experiment should be done to validate the hypothesis that fine-tuning starts from a near-optimal point.
>
> We agree with the reviewer that the concept of “near local optimum” is vaguely defined. However, we believe this is besides the point.
>
> The core assumption for deriving dimension-independent bounds in our theoretical results (decaying restricted Lipschitz coefficients) can be stated informally as “gradients evaluated at where optimization could be relevant is near low-dimensional.” This condition is also what we intended to verify in our experiments, and the empirical evidence suggests that gradients evaluated along optimization trajectories are indeed near low-dimensional (see Figure 2). We used the phrase "gradients evaluate near a local optimum" to really mean "gradients evaluated at where optimization could be relevant," since the former phrase is more succinct.
>
> We mildly agree with the reviewer that the wording “near local optimum” could become a point of confusion. We will further clarify in the main draft.
>
> [1] Wu, Y., Ren, M., Liao, R., & Grosse, R. (2018). Understanding short-horizon bias in stochastic meta-optimization. International Conference on Learning Representations.
>
> [2] Martens, J. (2020). New insights and perspectives on the natural gradient method. The Journal of Machine Learning Research, 21(1), 5776-5851.
>
> [3] Schaul, T., Zhang, S., & LeCun, Y. (2013, May). No more pesky learning rates. In International conference on machine learning (pp. 343-351). PMLR.
>
> [4] Zhang, G., Li, L., Nado, Z., Martens, J., Sachdeva, S., Dahl, G., ... & Grosse, R. B. (2019). Which algorithmic choices matter at which batch sizes? insights from a noisy quadratic model. Advances in neural information processing systems, 32.

---

### Official Review · Reviewer_vt7U · 2022-07-12

**Rating:** 8
**Confidence:** 3
**Soundness:** 4 excellent
**Presentation:** 4 excellent
**Contribution:** 4 excellent

**Summary:**


The paper considers differential privacy convex optimization in high-dimensional settings. Early theoretical results on private convex optimization say that empirical and population risk depends on the model size. However, empirical observations showed that the error could be much smaller in practice. Further theoretical work showed that if the gradients of the underlying loss function lie in a fixed low-rank space, then the risk can be bounded by its rank.

This paper is motivated by the empirical observation that given a pre-trained model, then using private gradient descent to fine-tune the model has good performance independent of model size. In a sense, this empirical observation contradicts theoretical results that state that differentially private convex learning degrades with model size.

The paper provides bounds independent of the model size on the empirical and population risk of private convex optimization. This result extends from prior work [JT14, STT20] that assumes that gradients belong to a fixed low-rank space. Furthermore, other results require learning the gradients subspace to get improved results or using public data to approximate the gradient subspace. Finally, the paper analyzes DP stochastic gradient descent without modification and only requires a condition called restricted Lipschitz continuity.

The analysis follows by partitioning the solution space into different orthogonal sub-spaces and then decomposing the gradient into a sum such that each term falls into a subspace. By the restricted Lipschitz continuity condition, each gradient term can be bounded by a constant.


**Questions:**

Could more intuition be provided about the is the effect of regularization in the analysis?


**Limitations:**

Yes

**Strengths And Weaknesses:**


The paper analyzes a practical algorithm (DP-SGD) for convex optimization and provides improved risk bounds under certain conditions that are more general than prior work. In addition, the bounds provided recover bounds from prior work that requires a low gradient subspace.
The idea of using restricted Lipschitz continuity conditions is new to my knowledge, and the analysis that follows is novel. Furthermore, experiments validate the theoretical claims.

The paper also motivates the theory in the convex setting to justify the high performance of pre-trained models in a non-convex setting. The paper claims and provides evidence that since pre-trained models are already close to a local optimum, they could lie in a small subspace. This argument gives some optimism about the future of private training large models.

---

> ### Author Response · Authors · 2022-08-02
> **Response to reviewer vt7U**
>
> We address a specific question raised by the reviewer.
>
> > Could more intuition be provided about the is the effect of regularization in the analysis?
>
> We thank the reviewer for their insightful question. The use of regularization yields the strongly convex objective $F_\alpha$ and enables us to leverage standard proof techniques in online convex optimization.
>
> Intuitively, regularization prevents the current iterate from deviating away from the initial iterate and yields a bound on this distance which does not grow with time. More technically, in Theorem 3.3, the regularization hyper-parameter $\alpha$ balances a term that depends on the distance to the optimum $D = \| x^{(0)} - x^* \|_2$ against a term that depends on the norms of gradients $\| \nabla F(x^{(t)}) \|_2$ when bounding the excess empirical loss. By carefully choosing the value for $\alpha$, one achieves an optimal tradeoff.
>
> Note under more restricted conditions (e.g., strictly low-rank gradients), one can obtain dimension-independent bounds without weight decay regularization [SSTT21]. Whether one can obtain these bounds without weight decay and under the more general condition (full rank, but “near” low-rank gradients) is an interesting open problem.

---

### Author Response · Authors · 2022-08-02
**Addressing questions**

We thank all reviewers for their detailed feedback and comments. Below, we address the concerns and questions of each reviewer separately.

---

### Author Response · Authors · 2022-08-02
**new revision that addresses questions from reviewers**

We thank the reviewers again for their time in reading our draft and providing detailed feedback. We have uploaded a new revision of our submission to reflect changes made based on reviewers' suggestions.

In particular, in Appendix D.2, Figures 6 and 7 contain results for Roberta-base and Roberta-large. For Roberta-large, due to the huge memory demand, we report results with 100 singular values in this revision. We are running experiments that compute thousands of singular values for this large model and will upload a further revision when results are ready.

Notably, for both Roberta-base and Roberta-large, we observe similar behaviours as for the DistilRoberta case (i.e., retraining in subspace recovers performance and singular values decay rapidly).

---

### Meta-Review · Area_Chair_ciLy · 2022-08-25

**Recommendation:** Accept
**Confidence:** Certain

**Metareview:**

The paper considers DP convex optimization in high-dimension, providing bounds independent of the model size on the empirical and population risk (extending prior work that assumes that gradients belong to a fixed low-rank space). All the reviewers agree that this is an important problem and the results are interesting, and support accepting this paper.

**Award:**

No

---

### Decision · Program_Chairs · 2022-09-14

Accept